# Magnetic Properties of 2D Nanowire Arrays: Computer Simulations

**DOI:** 10.3390/ma16093425

**Published:** 2023-04-27

**Authors:** Sergey V. Belim, Igor V. Bychkov

**Affiliations:** 1Department of Physics, Omsk State Technical University, 644050 Omsk, Russia; 2Department of Radiophysics and Electronics, Chelyabinsk State University, 454001 Chelyabinsk, Russia; bychkov@csu.ru

**Keywords:** nanowires 2D array, superantiferromagnetic ordering, phase transition

## Abstract

The paper considers a nanowires 2D array located in the nodes of a square lattice. Computer simulations use the Heisenberg model and Metropolis algorithm. The array consists of small nanowires that are monodomain. The exchange interaction orders the spins within a single nanowire. Dipole–dipole forces act between neighboring nanowires. The shape of an individual nanowire affects its magnetic anisotropy. Computer simulations examine the phase transition temperature and magnetization behavior of the system. The type of magnetic moments ordering in the array of nanowires depends on the orientation of their long axis. We consider two types of systems. The nanowires’ long axes are oriented perpendicular to the plane of their location in the first case. A dipole–dipole interaction results in first-type superantiferromagnetic ordering of the nanowires’ magnetic moments for such orientation. The nanowires’ long axes are oriented in the plane of the system in the second case. Dipole–dipole interaction results in second-type superantiferromagnetic ordering in such systems. The dependence of the phase transition temperature on the dipole–dipole interaction intensity is investigated.

## 1. Introduction

There has been significant interest in one-dimensional nanostructures (nanowires, nanocolumn, and nanorods) in recent times. Nanowires have a large length-to-diameter ratio. As a consequence, the surface-to-volume ratio is large for these objects. This shape gives nanowires unique electrical, optical and magnetic characteristics [1,2,3,4]. The magnetic properties of nanowires are determined by the anisotropy of their shape. The anisotropy axis is directed along the long axis of the nanowire. The anisotropy parameter is quite large due to the aspect ratio.

Models that describe the magnetic properties of an individual nanowire have made substantial progress. The characteristics and magnetic properties of magnetic nanowires arrays remain an open problem. For example, the magnetization and influence of the easy axis remain unexplored. Arrays of ferromagnetic nanowires have new collective physical properties.

Arrays of magnetic nanowires can be synthesized by electrochemical deposition in nanowires [5,6,7,8,9,10,11,12]. A well-defined pore architecture makes it possible to obtain arrays of nanowires with the necessary aspect ratio, composition, structure and density. There are also other technologies for obtaining arrays of nanowires: electron beam lithography [13] and interference lithography [14,15,16,17,18].

The static magnetic properties of the nanowire array are intensively studied using magnetometry methods [19,20,21]. Methods of ferromagnetic resonance [22,23,24] and Brilluin scattering [25,26] are used to study their dynamic properties. The effect of giant magnetic resistance is observed in multilayer films based on arrays of ferromagnetic nanowires. Therefore, they can be used in spintronic devices [27].

The existence of the anisotropy axis leads to the fact that the array of nanowires, in its properties, should be similar to the Ising model. The distance between the nanowires is large enough for exchange forces. Magnetostatic interaction by dipolar forces contributes most to the collective behavior of the system. Dipole–dipole interaction has various effects on processes in magnetic nanowires. Dipolar forces can reduce the coercivity of a nanowires array. These forces can influence the distribution of magnetization within an individual nanowire. The problem of interaction between nanowires requires separate consideration [28,29]. These interactions are studied experimentally by magnetometry using the example of Ni nanowires [30]. Dipolar forces can lead to supermagnetic ordering in a nanowires array. As shown by the simulation of ferromagnetic nanoparticle ensembles [31,32,33], different types of ordering are possible for the magnetic moments of the particles. The type of ordering depends on the orientation of the anisotropy direction relative to the substrate. It is possible that both the superferromagnetic phase and several types of superantiferromagnetic order exist. The direction of anisotropy for nanowires coincides with their long axis, so the type of ordering will be determined by the geometric location of the nanowires on the substrate.

Computer simulations of magnetic behavior for nanowires are preferably performed based on the micromagnetic modeling. There are several open source packages [34,35,36] and commercial products [37,38]. As a rule, the ellipsoidal model is used for micromagnetic modeling [39]. However, this model showed its limitations for single-domain particles with a large aspect ratio. Most models are adapted to experimental results by dependence of coercive force on the direction of the magnetic field [40,41,42].

The purpose of this article is computer modeling of the collective ordering of magnetic moments of ferromagnetic nanowires at their different orientation relative to the substrate.

## 2. Model and Computer Simulation

We consider an ordered 2D array of ferromagnetic nanowires. Nanowires are located in the nodes of a square lattice. The plane of the nanowires corresponds to the substrate. The distance between neighboring nanowires is d. Each nanowire has a square section with dimensions of one a×a×b, which spin the nanowire (b>a). The orientation of the nanowire in space is determined by the orientation of its long side b. We are investigating two system configurations. The nanowires are oriented perpendicular to the plane of the substrate in the first case. The nanowires are oriented in the plane of the substrate in the second case. The substrate surface is located in the OXY plane. The nanowires are oriented along the OZ axis in the first case. Nanowires are oriented along the OX axis in the second case. The system configurations are shown in Figure 1.

We use the Heisenberg model to describe the magnetic behavior of the system. The spin vector is mapped to each atom ${\vec{S}}_{i}=\left(S_{ix},S_{iy},S_{iz}\right)$, $\left|{\vec{S}}_{i}\right|=1/2$. Nanowires are made of ferromagnetic material. The exchange interaction orders the spins within a single nanowire. The interaction between the spins of neighbor nanowires is determined only by dipole–dipole forces. Dipole–dipole interaction is present between all spins. The dipole–dipole interaction in a single nanowire is accounted for as a demagnetizing factor. Between the spins for neighbor nanowires, the dipolar interaction is calculated directly. The Hamiltonian of the system includes two terms.
(1)$$H=-\underset{\langle i,j\rangle }{\sum }\left(J_{0}{\vec{S}}_{i}{\vec{S}}_{j}+A({\vec{S}}_{i})\right)+\underset{\left\{i,j\right\}}{\sum }J_{dd}({\vec{S}}_{i},{\vec{S}}_{j}).$$

${\vec{S}}_{i}$ is the spin in node number i. $J_{0}$ is the exchange integral. In the first term, the sum is calculated only from the nearest adjacent spins in one nanowire 〈i, j〉. This term takes into account the anisotropy of the form for the nanowire $A({\vec{S}}_{i})$. The second term describes the dipole–dipole interaction between spins $\left\{i,\;j\right\}$ from different nanowires.

The magnetization anisotropy is a consequence of the nanowires shape. The anisotropy parameter depends on the geometric dimensions of the nanowires. The presence of the form’s anisotropy is due to the long-range dipole–dipole interaction inside the nanowire. The anisotropy of the shape for the nanowire can be represented through demagnetizing factors $N_{x},\;N_{y},\;N_{z}$.
(2)$$\begin{array}{c} A({\vec{S}}_{i})=N_{x}{\left(S_{ix}\right)}^{2}+N_{y}{\left(S_{iy}\right)}^{2}+N_{z}{\left(S_{iz}\right)}^{2}.\\ N_{x}+N_{y}+N_{z}=1 \end{array}$$

We use demagnetizing factors for a ferromagnetic ellipsoid with a long axis b and two identical short axes a (b>a).
(3)$$\begin{array}{c} N_{1}=\frac{1}{n^{2}-1}\left(\frac{n}{2\sqrt{n^{2}-1}}\cdot ln\left(\frac{n+\sqrt{n^{2}-1}}{n-\sqrt{n^{2}-1}}\right)-1\right),\\ N_{2}=\frac{n}{2\left(n^{2}-1\right)}\left(n-\frac{1}{2\sqrt{n^{2}-1}}\cdot ln\left(\frac{n+\sqrt{n^{2}-1}}{n-\sqrt{n^{2}-1}}\right)\right).\\ n=b/a. \end{array}$$

$N_{1}$ is the demagnetizing factor along the long axis of the ellipsoid. $N_{2}$ is a demagnetizing factor along the minor axes of the ellipsoid.

The anisotropy term is determined by the orientation of the nanowires relative to the substrate. In the first case, all nanowires are arranged perpendicular to the substrate. The anisotropy direction is the OZ axis in this case.
(4)$$N_{x}=N_{2},\;N_{y}=N_{2},\;N_{z}=N_{1}.$$

The anisotropy axis lies in the plane of the substrate in the second case. All nanowires lie in the plane of the substrate and are oriented in one direction. We choose the direction of the OX axis parallel to the orientation of the nanowires. The anisotropy axis is directed along the OX axis in this case.
(5)$$N_{x}=N_{1},\;N_{y}=N_{2},\;N_{z}=N_{2}.$$

The exchange interaction dominates the ordering of spins within nanowires. The distance between neighbor nanowires is commensurate with their dimensions or exceeds it. Exchange forces are small at such distances. Dipole–dipole forces dominate the interaction of two spins from neighbor nanowires. A nanowire cannot be considered as a single macrospin. Non-uniform spin orientations are possible within the nanowire for nonzero temperature. Consider nanowires at zero temperature. In this case, all spins are oriented equally inside one nanowire. The middle spin of the nanowires coincides with the spin of a single node within the nanowire. This limit case allows predicting possible types of ordering. The energy of the dipole–dipole interaction depends on the orientation of the spins ${\vec{S}}_{i}$ and ${\vec{S}}_{j}$ relative to the radius vector ${\vec{r}}_{ij}$ between them.
(6)$$J_{dd}({\vec{S}}_{i},{\vec{S}}_{j})=B\frac{\left({\vec{S}}_{i}{\vec{S}}_{j})r_{ij}^{2}-3({\vec{S}}_{i}{\vec{r}}_{ij})({\vec{S}}_{j}{\vec{r}}_{ij}\right)}{r_{ij}^{5}}.$$

${\vec{r}}_{ij}$ is the radius vector between the spins ${\vec{S}}_{i}$ and ${\vec{S}}_{j}$. B is the relative intensity of dipolar forces. The distance between neighbor nanowires is d. This distance is characteristic of dipole–dipole forces in the system. The energy of the dipolar interaction between neighbor nanowires is determined by this distance.
(7)$$J_{dd}({\vec{S}}_{i},{\vec{S}}_{j})=B\frac{({\vec{S}}_{i}{\vec{S}}_{j})-3S_{id}S_{jd}}{d^{3}}.$$

$S_{id}$ and $S_{jd}$ are projections of the spins ${\vec{S}}_{i}$ and ${\vec{S}}_{j}$ in the direction of the radius vector between these spins. Computer simulation uses relative values of system parameters. We introduce the relative magnitude of the dipole–dipole interaction between the nanowires.
(8)$$R=\frac{B}{J_{0}d^{3}}.$$

The energy of dipolar forces in relative units takes on a simpler form.
(9)$$J_{dd}({\vec{S}}_{i},{\vec{S}}_{j})=RJ_{0}\left(({\vec{S}}_{i}\cdot {\vec{S}}_{j})-3{\vec{S}}_{id}{\vec{S}}_{jd}\right).$$

The relative intensity R depends on the substance of the nanowires and the distance between them. The computer experiment uses different values of R. If the distance between the nanowires decreases, then the relative intensity R increases.

The dipole–dipole interaction energy has different effects on spin ordering. If the nanowires are oriented perpendicular to the substrate, the anisotropy axis is perpendicular to the radius vector between the spins. The spins are oriented mainly along the OZ axis. The first term modulo will dominate the second in Formula (9).
(10)$$|{\vec{S}}_{i}\cdot {\vec{S}}_{j}|\gg 3\left|S_{id}S_{jd}\right|.$$

The energy of dipolar forces will have a positive value at co-directional spins and a negative value at oppositely directed spins.
(11)$$J_{dd}({\vec{S}}_{i},{\vec{S}}_{j})\approx RJ_{0}({\vec{S}}_{i}\cdot {\vec{S}}_{j}).$$

In this case, it is energetically advantageous for the spins to navigate in one direction within one nanowire. For neighbor nanowires, the minimum energy is at the opposite orientation. Thus, superantiferromagnetic ordering is possible in the system. The orientation of the nanowire magnetizations is shown in Figure 2. Such ordering will be called the superantiferromagnetic phase of the first type.

If the nanowires are oriented parallel to the substrate, the spins are oriented substantially along the OX axis. In this case, two cases of dipolar interaction should be considered. If two neighbor nanowires are located along the OX axis, then the spin orientation is parallel to the radius vector between the particles. The second term in the dipole–dipole interaction energy is greater than the first.
(12)$$|{\vec{S}}_{i}\cdot {\vec{S}}_{j}|<3\left|S_{id}S_{jd}\right|.$$

The dipole–dipole interaction energy will be positive at the opposite spin orientation and negative for co-directed spins.
(13)$$J_{ddx}({\vec{S}}_{i},{\vec{S}}_{j})=-2RJ_{0}({\vec{S}}_{i}{\vec{S}}_{j}).$$

In this case, it is advantageous to orient the magnetic moments of the nanowires in one direction.

For two nanowires neighboring the OY axis, the anisotropy axis is perpendicular to the radius vector between them. The second term becomes negligible compared to the first. The dipole–dipole interaction energy is positive for equally oriented spins and negative for oppositely oriented spins.
(14)$$J_{ddy}({\vec{S}}_{i},{\vec{S}}_{j})=RJ_{0}({\vec{S}}_{i}\cdot {\vec{S}}_{j}).$$

In this case, ferromagnetic ordering occurs within one nanowire. The magnetic moments of nanowires with the same coordinate along the OX axis are co-directed. Magnetic moments of nanowires with the same coordinate along the OY axis are directed oppositely. Dipole–dipole interaction signs for different locations of nanowires are shown in Figure 3.

In this case, superantiferromagnetic ordering occurs in the system. The array of nanowires is divided into two sublattices with magnetic moments oriented oppositely. Both sublattices are a plurality of parallel chains of nanowires separated by distance 2d (Figure 4). Such ordering will be called the superantiferromagnetic phase of the second type.

The description of the phase transition is based on the order parameter. We enter three order parameters to study different types of ordering. First, we introduce the magnetization of an individual nanowire, such as the average spin of its atoms. The magnetic moment of the number $\left(l,\;k\right)$ nanowire is calculated as the coordinate sum of the spins included in it.
(15)$${\vec{m}}_{lk}=\left(m_{lkx},m_{lky},m_{lkz}\right)=\left(\underset{S_{i}\in m_{lk}}{\sum }{\vec{S}}_{i}\right)/\left(ba^{2}\right).$$

The total magnetization of the array is equal to the sum of the magnetic moments in the individual nanowires.
(16)$$\vec{m}=\overset{L}{\underset{l,k=1}{\sum }}{\vec{m}}_{lk}/L^{2}.$$

L is the number of nanowires along one axis.

The transition to the superantiferromagnetic state is described by the difference in magnetic moments of the sublattices. In antiferromagnetic ordering of the first type, it is necessary to calculate magnetization differences of a sublattice from nanowires with an even sum of indices and a sublattice with an odd sum of indices.
(17)$${\vec{m}}_{1}=\left(\overset{L}{\underset{\begin{array}{c} l,k=1\\ l+k=even \end{array}}{\sum }}{\vec{m}}_{lk}-\overset{L}{\underset{\begin{array}{c} l,k=1\\ l+k=odd \end{array}}{\sum }}{\vec{m}}_{lk}\right)/L^{2}.$$

Superantiferromagnetic ordering of the second type requires its own order parameter. The second order parameter is also calculated as the magnetization difference of the sublattices. The first sublattice includes nanowires with an even coordinate along the OX axis. The second sublattice consists of nanowires with an odd coordinate along the OX axis.
(18)$${\vec{m}}_{a}=\left(\underset{k=odd}{\sum }{\vec{m}}_{lk}-\underset{k=even}{\sum }{\vec{m}}_{lk}\right)/L^{2}.$$

The temperature for both types of superantiferromagnetic phase transition is determined on the basis of finite-dimensional scaling theory [43]. The properties of an infinite system are approximated based on the results for finite-size systems. Computer simulation calculates the parameters of systems for different linear sizes L×L. We used systems with a linear size from L=8a to L=16a in ΔL=2a increments. Metropolis’s algorithm forms the thermodynamic states for the system. The peculiarity of the algorithm for this system consists of a single bypass of spins in one nanowire. Adjacent nanowires affect the processed nanowire through an interspine interaction. The transition to the adjacent nanowire occurs after an attempt to order the spins of a single nanowire. We calculated $10^{6}$ Monte Carlo steps per spin. Order parameters are calculated for each state. Averaging by thermodynamic states allows us to determine the behavior of the thermodynamic parameters in the system depending on temperature. We compute thermodynamic functions for each order parameter. The behavior of thermodynamic functions during temperature changes characterizes the phase transition. The first computed thermodynamic functions are Binder cumulants [44].
(19)$$\begin{array}{c} U\left(L,T\right)=1-\frac{\langle m^{4}\rangle }{3\langle m^{2}\rangle {}^{2}},\;\;m^{2}=m_{x}^{2}+m_{y}^{2}+m_{z}^{2}.\\ U_{1}\left(L,T\right)=1-\frac{\langle m_{1}^{4}\rangle }{3\langle m_{1}^{2}\rangle {}^{2}},\;\;m_{1}^{2}=m_{1x}^{2}+m_{1y}^{2}+m_{1z}^{2},\\ U_{2}\left(L,T\right)=1-\frac{\langle m_{2}^{4}\rangle }{3\langle m_{2}^{2}\rangle {}^{2}},\;\;m_{2}^{2}=m_{2x}^{2}+m_{2y}^{2}+m_{2z}^{2}.\; \end{array}$$

The dependence of Binder cumulants on temperature allows us to determine the phase transition point. The Binder cumulant is 2/3 in the ordered phase and decreases in transition to the disordered phase with increasing temperature. The rate of decline for Binder cumulants depends on the size of the system. The Binder cumulants do not depend on the size of the system at the phase transition point. It follows that the Binder cumulant dependency plots intersect at one point. This point corresponds to the phase transition temperature.

We also compute a fourth-order energy cumulant.
(20)$$V=1-\frac{\langle E^{4}\rangle }{3\langle E^{2}\rangle {}^{2}}.$$

E is the energy of the system calculated from Hamiltonian (1). This cumulant also allows us to determine the phase transition temperature. At the critical point, there is a pronounced minimum on the cumulant–temperature plot. The energy cumulant also allows us to determine the genus of phase transition.

Averaging over the thermodynamic states of the system makes it possible to calculate the magnetic susceptibility of the system.
(21)$$\begin{array}{c} \chi =\frac{\partial m}{\partial h}=L^{2}\left(\langle m^{2}\rangle -\langle m\rangle {}^{2}\right)/T.\\ \chi_{1}=\frac{\partial m_{1}}{\partial h}=L^{2}\left(\langle m_{1}^{2}\rangle -\langle m_{1}\rangle {}^{2}\right)/T,\\ \chi_{2}=\frac{\partial m_{2}}{\partial h}=L^{2}\left(\langle m_{2}^{2}\rangle -\langle m_{2}\rangle {}^{2}\right)/T.\; \end{array}$$

Magnetic susceptibility for antiferromagnetic order parameters is equal to the difference of magnetic susceptibility for sublattices. For phase transitions of the second kind, magnetic susceptibilities experience a jump in the phase transition point. With each type of ordering, only one of these susceptibilities exhibits abnormal behavior. A comparison of susceptibility plots allows us to find out what type of ordering occurred in the system.

We also calculate the heat capacity of the system from the average energy for the system.
(22)$$C=\frac{\partial E}{\partial T}=L^{2}\left(\langle E^{2}\rangle -\langle E\rangle {}^{2}\right)/T.$$

The heat capacity plot also experiences a jump at the phase transition point.

## 3. Results

A computer experiment simulates an array of nanowires with linear dimensions a=2 and b=16. The demagnetization coefficients in the anisotropy parameter for such nanowires are $N_{1}=0.028$ and $N_{2}=0.486$. The relative intensity of the dipole–dipole interaction varies from R=0.2 to R=0.7 in steps ΔR=0.1 in a computer experiment. The distance between the nanowires is not explicitly given because it enters the dipolar force intensity value R.

An array of nanowires perpendicular to the substrate is examined in a first computer experiment. The dependence of three order parameters on the temperature for R=0.5 is shown in Figure 5.

The plot in Figure 5 shows the superantiferromagnetic ordering of the first type in the system. The dependence of magnetic susceptibility $\chi_{1}$ on temperature supports this conclusion (Figure 6a). A plot of heat capacity versus temperature is shown in Figure 6b. The heat capacity graph has a stepped form, which is characteristic of the second-order phase transitions. The peak in magnetic susceptibility increases with the size of the system. The heat capacity plot changes little as the system increases in size. This pattern is characteristic of the two-dimensional Ising model. We use the Heisenberg model with the anisotropy axis. A computer experiment shows that the critical behavior of the system for the phase transition under study is described by the Ising model.

The phase transition temperature $T_{1}$ with the greatest accuracy can be calculated based on Binder cumulants $U_{1}$. Two examples of Binder cumulants versus temperature for systems with different sizes at R=0.3 and R=0.7  are shown in Figure 7.

Binder cumulant plots have a fairly explicit intersection point for systems of various sizes. This fact indicates a phase transition of the second order. The order of the phase transitions can also be determined from the analysis of energy cumulants V. Examples of the dependence of energy cumulants on temperature are shown in Figure 8 for R=0.3 and R=0.7.

For second-order phase transitions, the plot of cumulant V versus $L^{-2}$ at the phase transition point is linear and intersects the ordinate axis at 2/3 (Murtazaev). The corresponding plots are shown in Figure 9.

As can be seen from Figure 9, the superantiferromagnetic phase transition of the first type is a second-order phase transition. Calculations for all R values give the same results. A plot of phase transition temperature versus dipole–dipole interaction intensity R is shown in Figure 10.

The phase transition temperature $T_{1}$ does not increase linearly with the increasing dipole–dipole interaction intensity R. The dependence of the phase transition temperature *T*_1_ on the intensity of dipolar forces *R* is approximated by the logarithmic law with high accuracy. A plot of $T_{1}$ versus lnR is shown in Figure 11.

The law for dependence of the phase transition temperature into the first-type superantiferromagnetic phase is written based on the linear approximation of the graph in Figure 11.
(23)$$T_{1}=(0.31\pm 0.01)\ln R+(1.41\pm 0.01).$$

A second computer experiment examines an array of nanoparticles with the anisotropy direction along the OX axis. The dependence of the three order parameters on the temperature in this computer simulation is shown in Figure 12 at R=0.5.

Superantiferromagnetic ordering of the second type occurs in this system. The phase transition in the system is confirmed by the dependencies of magnetic susceptibility $\chi_{2}$ and heat capacity C on temperature (Figure 13).

Thermodynamic functions show behavior characteristic of the Ising model as well as for superantiferromagnetic ordering of the first type. Binder cumulants $U_{2}$ are used to more accurately determine the phase transition temperature $T_{2}$. Two examples of Binder cumulants $U_{2}$ dependence on temperature T at R=0.3 and R=0.7 are shown in Figure 14.

Binder cumulants have an explicit intersection point characteristic for second-order phase transitions. Energy cumulants V are calculated to confirm the second order of the phase transition. Examples of the energy cumulants V dependence on temperature T are shown in Figure 15.

The plots of cumulant V vs. $L^{-2}$ at the phase transition point are linear (Figure 16). The point of intersection of the plot with the ordinate axis is close to 2/3. These two facts support the second-order phase transition [45]. Other values R give the same results.

A plot of the phase transition temperature $T_{2}$ versus the dipole–dipole interaction intensity R is shown in Figure 17.

The phase transition temperature $T_{2}$ increases nonlinearly with increasing dipolar forces. The logarithmic function is the best approximation of this dependence. A plot of $T_{2}$ versus lnR is shown in Figure 18.

A linear approximation of plot 18 allows the function coefficients to be determined.
(24)$$T_{2}=\left(0.232\pm 0.002\right)\ln R+\left(1.134\pm 0.003\right).$$

The temperature for the second type of superantiferromagnetic ordering is lower than the first due to the total thickness of the system. The array of perpendicular nanowires creates a film with a greater thickness than the array of nanowires in the plane of the substrate.

## 4. Conclusions

We performed computer simulations of a 2D array of dipole–dipole interaction nanowires. Dipolar forces can cause superantiferromagnetic ordering in the system. Two types of antiferromagnetic ordering are possible in a nanowires array. The orientation of the nanowires relative to the substrate determines the type of ordering. If the nanowires are located perpendicular to the substrate, then their magnetization forms two square sublattices embedded in each other. Magnetic moments of sublattices are oriented oppositely. If the nanowires are parallel to the substrate, then the array is divided into chains. Nanowires in one chain have magnetic moments oriented in one direction. Magnetic moments in neighbor chains are oriented oppositely. The reason for different types of ordering is the anisotropy of the nanowires shape. Anisotropy of the form is the cause of nanowires magnetic anisotropy. The orientation of the nanowire anisotropy axis determines the type of superantiferromagnetic ordering.

Both types of superantiferromagnetic ordering are realized as a phase transition of the second kind. Thermodynamic functions near the phase transition point exhibit behavior as in the two-dimensional Ising model. The Ising model describes systems with an easy magnetization axis. In a 2D array of nanowires, the easy magnetization axis coincides with the long axis of the nanowires. The phase transition temperature for both types of ordering depends on the dipole–dipole interaction intensity by logarithmic law. The coefficients for each type of superantiferromagnetic transition have a different meaning.

The magnetic properties of nanowires-based metamaterials play an important role in their applied use [46,47]. Arrays of nanowires are a promising material for anodes in lithium-ion batteries [48,49]. Information about phase transitions in these metamaterials is important for device design. The differences in the magnetic properties of arrays of Co and Ni nanowires with different orientations were investigated on the basis of their magnetization [50]. The results of this work are consistent with our conclusions. We considered two limiting cases of nanowire orientation. Arrays of nanowires angled to the substrate or having a thickness gradient [51] may exhibit additional supermagnetic ordering phases.

## Figures and Tables

**Figure 1 materials-16-03425-f001:**
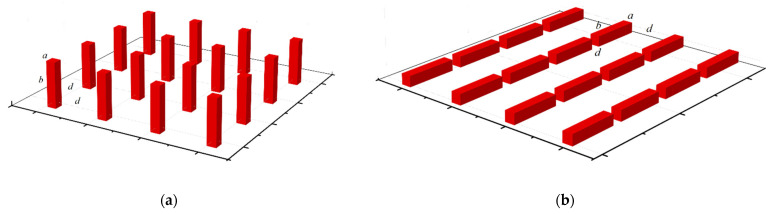
The geometry of the arrangement of the nanowires for two cases: (**a**) orientation along the OZ axis, (**b**) orientation along the OX axis.

**Figure 2 materials-16-03425-f002:**
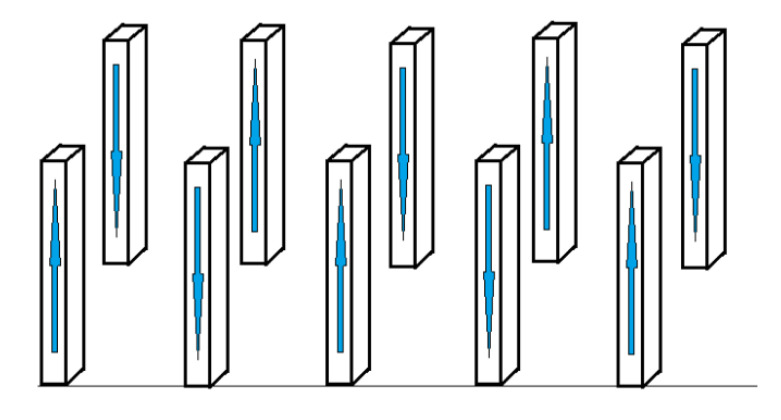
Superantiferromagnetic ordering in an array of nanowires perpendicular to the substrate.

**Figure 3 materials-16-03425-f003:**
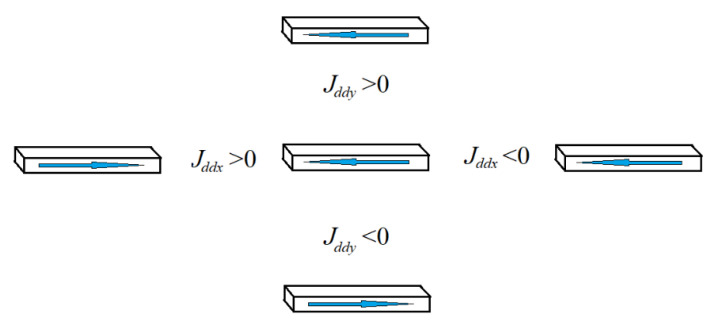
Dipole–dipole interaction energy at different magnetization orientation of nanowires.

**Figure 4 materials-16-03425-f004:**
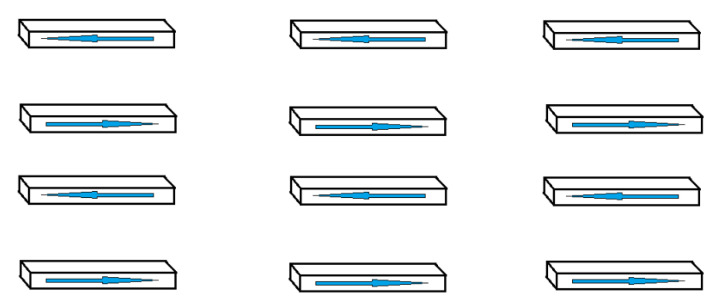
Chains of particles ordered by dipole–dipole interaction.

**Figure 5 materials-16-03425-f005:**
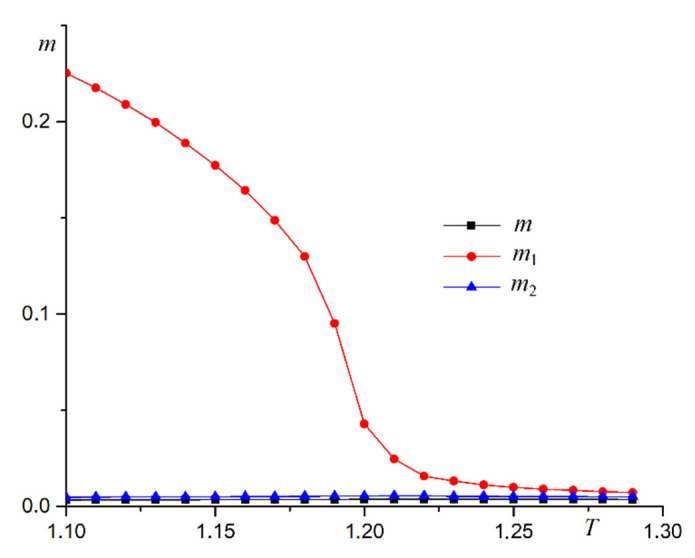
Dependence of three order parameters on temperature at R=0.5 for nanowires perpendicular to the substrate (magnetization m, antiferromagnetic order parameter of the first type $m_{1}$, antiferromagnetic order parameter of the second type $m_{2}$).

**Figure 6 materials-16-03425-f006:**
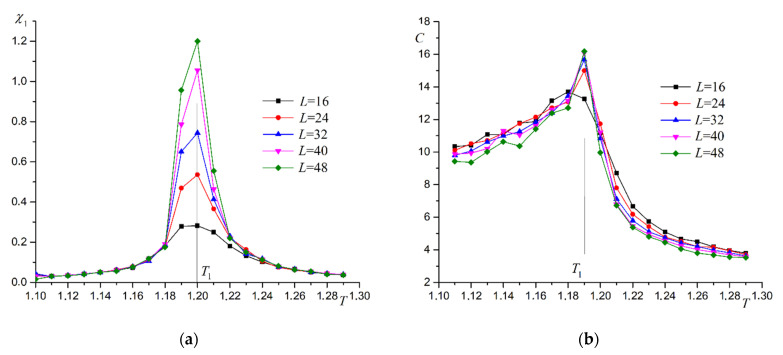
Dependence of magnetic susceptibility $\chi_{1}$ (**a**) and heat capacity C (**b**) on temperature T at R=0.5.

**Figure 7 materials-16-03425-f007:**
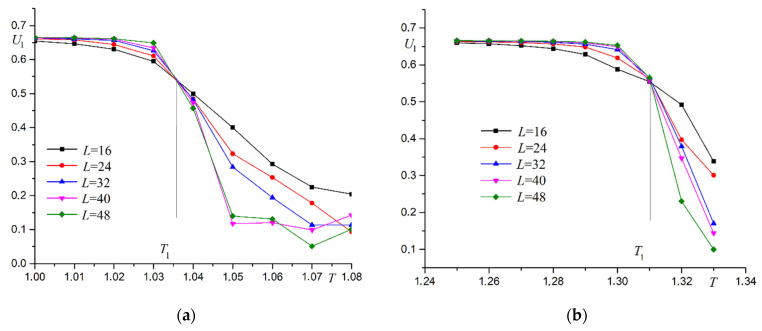
Dependence of Binder cumulants $U_{1}$ on temperature T (**a**) R=0.3, (**b**) R=0.7.

**Figure 8 materials-16-03425-f008:**
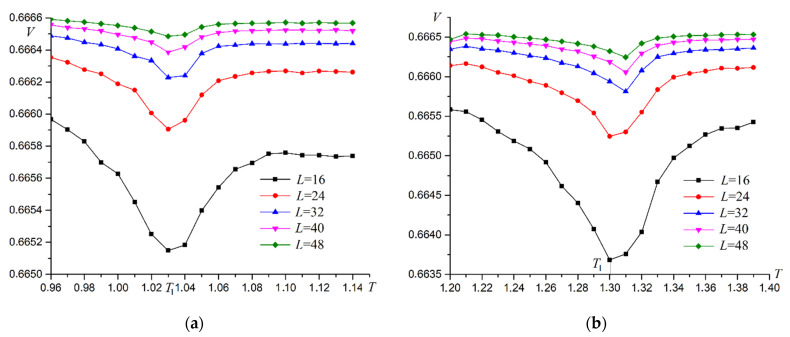
Dependence of energy cumulants V on temperature T. (**a**) R=0.3, (**b**) R=0.7.

**Figure 9 materials-16-03425-f009:**
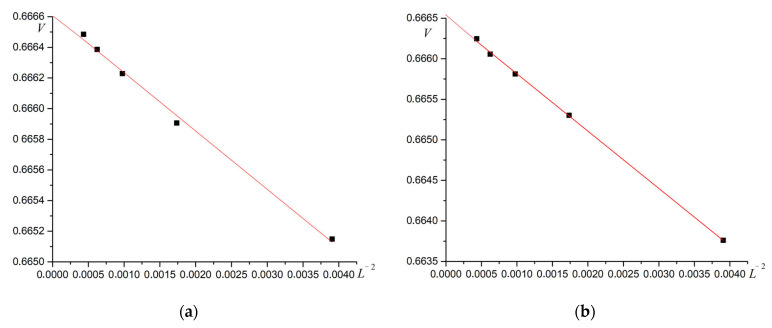
Dependence of energy cumulants V on $L^{-2}$ at the phase transition point. (**a**) R=0.3, (**b**) R=0.7.

**Figure 10 materials-16-03425-f010:**
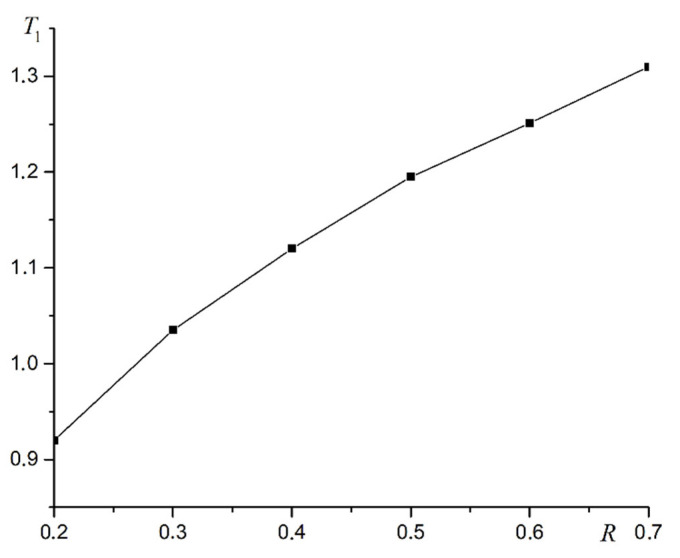
Plot of the temperature for the first type superantiferromagnetic transition $T_{1}$ versus the intensity of the dipole–dipole interaction R.

**Figure 11 materials-16-03425-f011:**
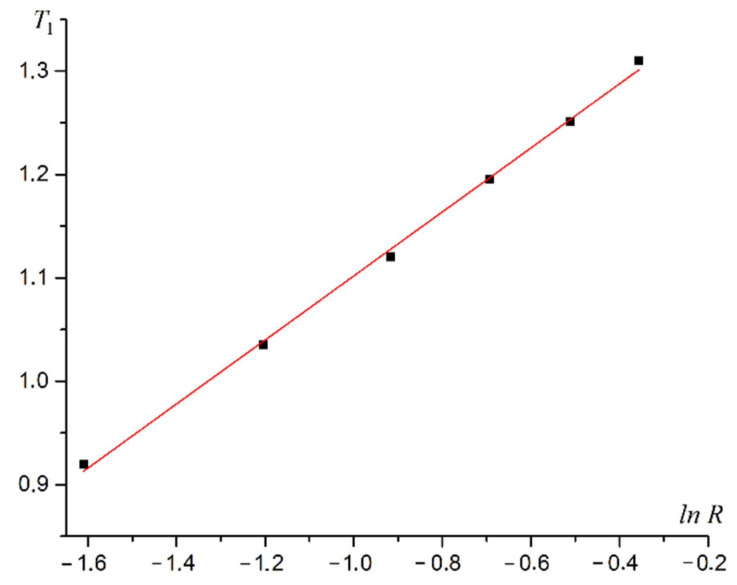
A plot of $T_{1}$ versus lnR.

**Figure 12 materials-16-03425-f012:**
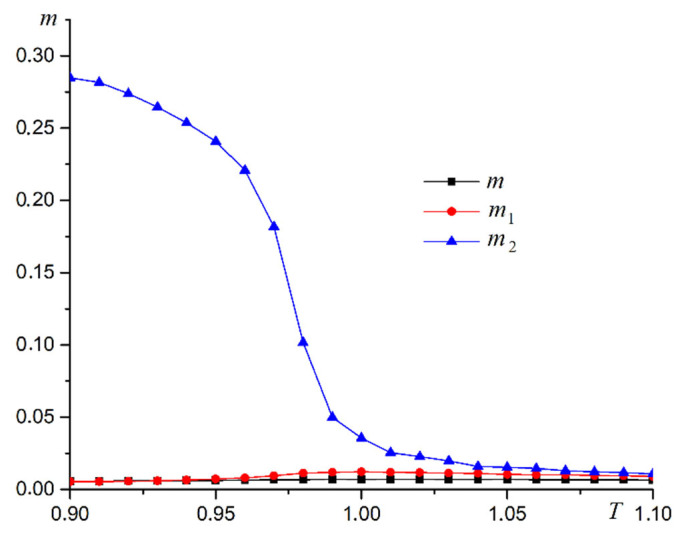
Dependence of three order parameters on temperature for nanowires parallel to substrate at R=0.5. (magnetization m, antiferromagnetic order parameter of the first type $m_{1}$, antiferromagnetic order parameter of the second type $m_{2}$).

**Figure 13 materials-16-03425-f013:**
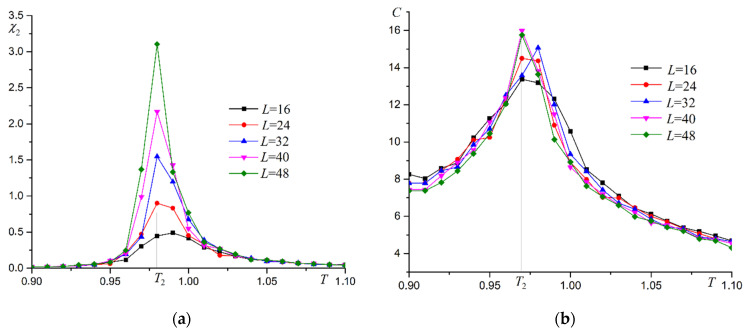
Dependence of magnetic susceptibility $\chi_{2}$ (**a**) and heat capacity C (**b**) on temperature T at R=0.5.

**Figure 14 materials-16-03425-f014:**
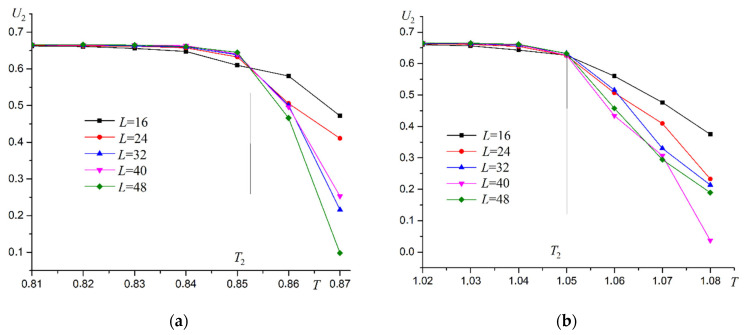
Dependence of Binder cumulants $U_{2}$ on temperature T. (**a**) R=0.3, (**b**) R=0.7.

**Figure 15 materials-16-03425-f015:**
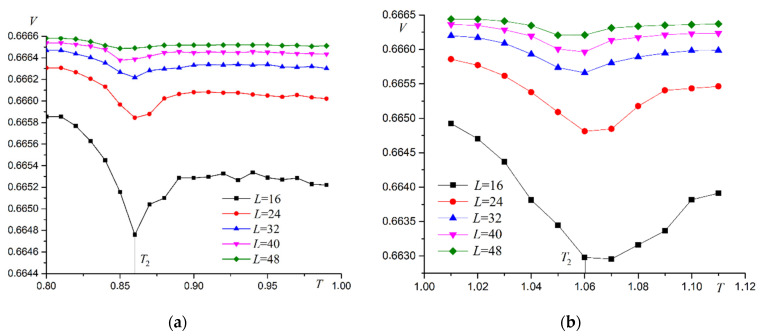
Dependence of energy cumulants V on temperature T for second type superantiferromagnetic transition. (**a**) R=0.3, (**b**) R=0.7.

**Figure 16 materials-16-03425-f016:**
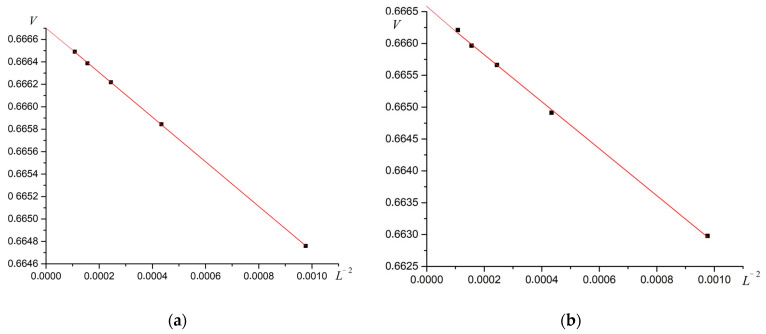
Dependence of energy cumulants V on $L^{-2}$ at the point of second-type superantiferromagnetic phase transition. (**a**) R=0.3, (**b**) R=0.7.

**Figure 17 materials-16-03425-f017:**
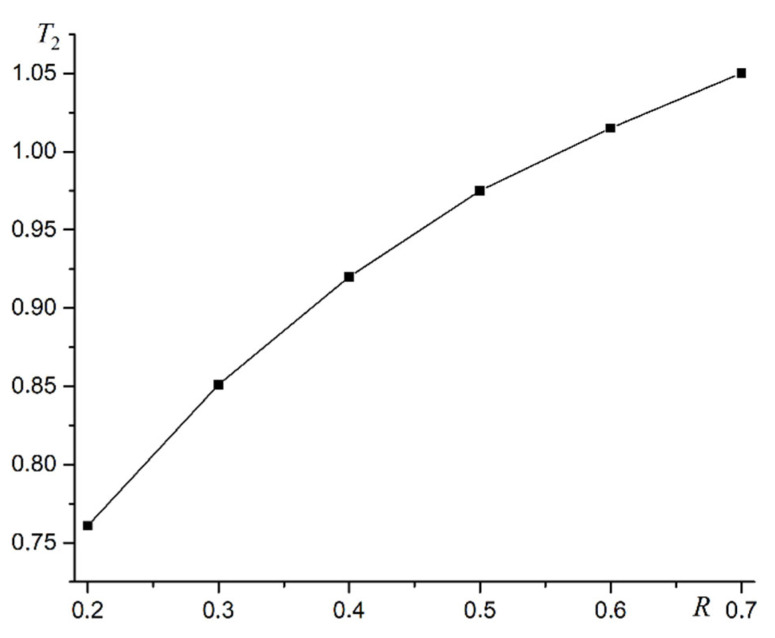
Plot for the temperature $T_{2}$ of the second-type superantiferromagnetic transition versus the intensity of the dipole–dipole interaction R.

**Figure 18 materials-16-03425-f018:**
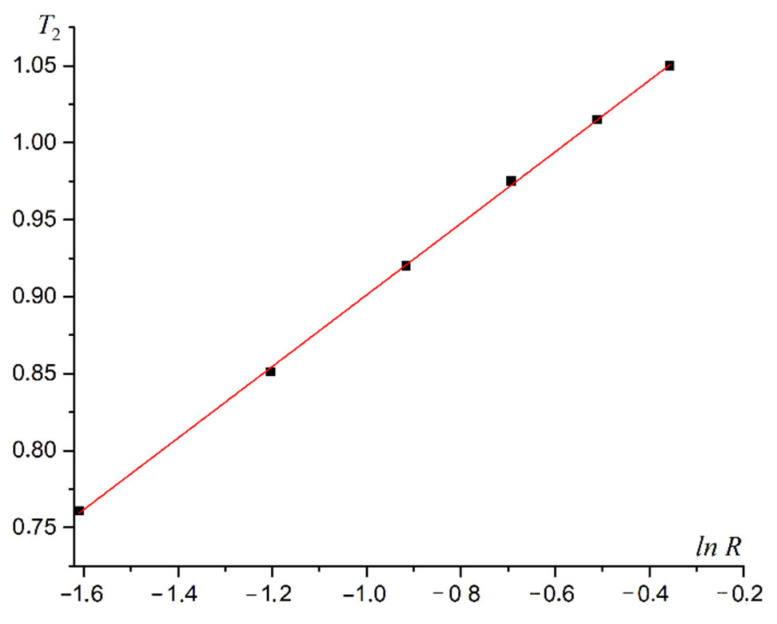
Plot of the temperature $T_{2}$ of the second-type superantiferromagnetic transition versus the intensity of dipole–dipole interaction lnR.

## Data Availability

Not applicable.

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
