# Peer review of "Magnetic Properties of 2D Nanowire Arrays: Computer Simulations"

_materials, 2023, doi:10.3390/ma16093425_

Round 1
Reviewer 1 Report
This manuscript clarified magnetic ordering and phase transition behavior in 2D nanowire arrays by using Heisenberg model and Metropolis algorithm. Shape anisotropy, exchange interaction and dipole-dipole interaction were included in their model and the effect of the relative magnitude of dipole-dipole interaction was investigated. They also show the effect of the orientation, that is shape anisotropy, of the nanowires. They demonstrated that phase transition occurs at lower temperature when the relative interaction is smaller. The transition was shown to be second order. The computational methodology and results are scientifically sound. Overall, I would recommend acceptance of this manuscript. I have only minor comments listed below:
1. Line 83 to 84, the authors said, “The spin vector is mapped to each atom Si = …” and “A nanowire cannot be considered as a single macrospin.” in line 114 to 115 but the author also said “Si is the spin in node number i.” in line 88. In addition, the method to discretize the magnetic spin in one nanowire cannot be found in the manuscript. I guess one spin was assigned to one node but if not, how it was modelled should be clearly explained in the manuscript.
2. Fig. 1 (b), the leftmost nanowire is aligned to the grid of left side but the rightmost one is aligned to the grid of right side. I guess it should be aligned to the grid of the left side.
3. Eq. (2), the superscripts of x, y and z should be subscripts according to line 84.
4. Line 189, the authors said they use Metropolis’s algorithm but the details of the methodology cannot be found in the manuscript. It should be explained for reproducibility.
5. The Eq. (23) next to line 312 should be Eq. (24) since there is Eq. (23) next to line 273. The authors determined a linear approximation of the dependence of transition temperature on the relative interaction and the function is different when the orientation is different. If they add some comments on the meaning or mechanism of it, it would be of interest to the readers.
Author Response
Dear reviewer!
We thank you for your constructive comments. We provide answers to your questions and comments.
- Line 83 to 84, the authors said, “The spin vector is mapped to each atom Si = …” and “A nanowire cannot be considered as a single macrospin.” in line 114 to 115 but the author also said “Si is the spin in node number i.” in line 88. In addition, the method to discretize the magnetic spin in one nanowire cannot be found in the manuscript. I guess one spin was assigned to one node but if not, how it was modelled should be clearly explained in the manuscript.
Answer: A paragraph has been added to the text of the article:
“Non-uniform spin orientations are possible within the nanowire for nonzero temperature. Consider nanowires at zero temperature. In this case, all spins are oriented equally inside one nanowire. The middle spin of the nanowires coincides with the spin of a single node within the nanowire. This limit case allows to predict possible types of ordering.”
- Fig. 1 (b), the leftmost nanowire is aligned to the grid of left side but the rightmost one is aligned to the grid of right side. I guess it should be aligned to the grid of the left side.
Answer: Picture is changed.
- Eq. (2), the superscripts of x, y and z should be subscripts according to line 84.
Answer: Equation (2) is changed
- Line 189, the authors said they use Metropolis’s algorithm but the details of the methodology cannot be found in the manuscript. It should be explained for reproducibility.
Answer: A paragraph has been added to the text of the article:
“Metropolis's algorithm forms the thermodynamic states for the system. The peculiarity of the algorithm for this system consists in a single bypass of spins in one nanowire. Adjacent nanowires affect the processed nanowire through an interspine interaction. The transition to the adjacent nanowire occurs after an attempt to order the spins of a single nanowire. We calculated 10^6 Monte-Carlo steps per spin..”
- The Eq. (23) next to line 312 should be Eq. (24) since there is Eq. (23) next to line 273. The authors determined a linear approximation of the dependence of transition temperature on the relative interaction and the function is different when the orientation is different. If they add some comments on the meaning or mechanism of it, it would be of interest to the readers.
Answer: We corrected the equation number.
We added a paragraph to the text of the article.
“The temperature for the second type of superantiferromagnetic ordering is lower than the first due to the total thickness of the system. The array of perpendicular nanowires creates a film with a greater thickness than the array of nanowires in the plane of the substrate.”
Reviewer 2 Report
1.The topic is suitable for this journal. It is of interest to the community.
2. This is a theoretical simulation based on classical theories, and the author should demonstrate the outstanding scientific significance and application value of this work. Therefore, it is suggested that this paper can provide some experimental data or other people's research reports that are consistent with the simulation results.
3. In this paper, the dipole-dipole effect should come from between adjacent nanowires. But it also points out that "The presence of the form's anisotropy is due to the long-range dipole-dipole interaction inside the nanowire", The author is asked to clarify this contradictory statement.
4. In Fig.2b, in addition to the vertical spacing d of nanowires, there is also a parallel spacing between nanowires. How does the author consider this when modeling? What is the basis for ignoring it in this article?
5. The demagnetization factor in this paper adopts the ellipsoid model. But is it more accurate to use cylindrical demagnetization factor for nanowires of finite length?
6. The simulation in this paper seems to consider only the dipole-dipole effects of the nearest nanowires. Do the dipole-dipole effects of second nearest neighbor and second nearest neighbor need to be considered? How much calculation error will be caused by ignoring these factors?
7. Eq. (23) is a linear formula, while the curves in Fig.18 are non-linear and do not coincide.
8. The thematic background of this manuscript needs to be further emphasized. Rcent literature, MATERIALS TODAY PHYSICS, 2023, 31, 100988, DOI: 10.1016/j.mtphys.2023.100988; ADVANCED OPTICAL MATERIALS, 2019, 7(19), 19, 1900689, DOI: 10.1002/adom.201900689.
Author Response
Dear reviewer!
We thank you for your constructive comments. We provide answers to your questions and comments.
- This is a theoretical simulation based on classical theories, and the author should demonstrate the outstanding scientific significance and application value of this work. Therefore, it is suggested that this paper can provide some experimental data or other people's research reports that are consistent with the simulation results.
Answer: We added a paragraph at the end of the conclusion.
“The magnetic properties of nanowires-based metamaterials play an important role in their applied use [46,47]. Arrays of nanowires are a promising material for anodes in lithium-ion batteries [48,49]. Information about phase transitions in these metamaterials is important for device design. The difference in magnetic properties of arrays Co, Ni nanowires with different orientations was investigated on the basis of their magnetization [50]. The results of this work are consistent with our conclusions. We considered two limiting cases of nanowire orientation. Arrays of nanowires angled to the substrate or having a thickness gradient [51,52] may exhibit additional supermagnetic ordering phases.”
- In this paper, the dipole-dipole effect should come from between adjacent nanowires. But it also points out that "The presence of the form's anisotropy is due to the long-range dipole-dipole interaction inside the nanowire", The author is asked to clarify this contradictory statement.
Answer: We corrected the text.
“The exchange interaction orders the spins within a single nanowire. The interaction between the spins of neighbor nanowires is determined only by dipole-dipole forces. Dipole-dipole interaction is present between all spins. The dipole-dipole interaction in a single nanowire is accounted for as a demagnetizing factor. Between the spins for neighbor nanowires, the dipolar interaction is calculated directly.”
- In Fig.2b, in addition to the vertical spacing d of nanowires, there is also a parallel spacing between nanowires. How does the author consider this when modeling? What is the basis for ignoring it in this article?
Answer: Parallel interaction between nanowires is considered. This interaction is automatically taken into account when calculating paired interactions between spins from neighbor nanowires. The distance is considered to be d.
- The demagnetization factor in this paper adopts the ellipsoid model. But is it more accurate to use cylindrical demagnetization factor for nanowires of finite length?
Answer: Demagnetizing factors for ellipsoid and cylinder nanowires differ little. This small difference is due to the shape of the nanowire. The difference in the values of the demagnetizing factors does not exceed the errors in the Monte Carlo method. But the shape of the ellipsoid is more convenient in computer modeling.
- The simulation in this paper seems to consider only the dipole-dipole effects of the nearest nanowires. Do the dipole-dipole effects of second nearest neighbor and second nearest neighbor need to be considered? How much calculation error will be caused by ignoring these factors?
Answer:
Dipolar forces rapidly decrease with distance. Taking into account the neighbors following the nearest ones leads to amendments in the second sign after the comma and does not affect the main results.
- Eq. (23) is a linear formula, while the curves in Fig.18 are non-linear and do not coincide.
Answer: We replaced Figure 18. In the text of the article, Figure 17 was erroneously inserted twice.
- The thematic background of this manuscript needs to be further emphasized. Rcent literature, MATERIALS TODAY PHYSICS, 2023, 31, 100988, DOI: 10.1016/j.mtphys.2023.100988; ADVANCED OPTICAL MATERIALS, 2019, 7(19), 19, 1900689, DOI: 10.1002/adom.201900689.
Answer: We've added links to the articles.
Reviewer 3 Report
In this article, the authors simulated a 2D array of dipole-dipole interaction nanowires. There are two types of possible antiferromagnetic ordering in this nanowires array. The results are reasonable. However, this work does not present much significance to the field of nanomaterials or nano magnetism. Hence, I do not recommend the publication on the Material journal.
Author Response
The magnetic properties of metamaterials based on an nanowires array have been actively investigated recently. These nanomaterials have great prospects for use in various devices. Information on possible types of super-ordering plays an important role in the design and operation of nanowire-based devices.